# Statistical Quantification of the Effects of Marker Misplacement and Soft-Tissue Artifact on Shoulder Kinematics and Kinetics

**DOI:** 10.3390/life12060819

**Published:** 2022-05-31

**Authors:** Maxence Lavaill, Saulo Martelli, Graham K. Kerr, Peter Pivonka

**Affiliations:** 1School of Mechanical, Medical and Process Engineering, Queensland University of Technology, Brisbane, QLD 4000, Australia; saulo.martelli@qut.edu.au (S.M.); peter.pivonka@qut.edu.au (P.P.); 2Queensland Unit for Advanced Shoulder Research, Brisbane, QLD 4000, Australia; g.kerr@qut.edu.au; 3Medical Device Research Institute, College of Science and Engineering, Flinders University, Tonsley, SA 5042, Australia; 4Movement Neuroscience Group, School of Exercise & Nutrition Sciences, Queensland University of Technology, Brisbane, QLD 4000, Australia

**Keywords:** skin markers, shoulder, kinematics, kinetics, error propagation, soft-tissue artifact, marker misplacement

## Abstract

The assessment of shoulder kinematics and kinetics are commonly undertaken biomechanically and clinically by using rigid-body models and experimental skin-marker trajectories. However, the accuracy of these trajectories is plagued by inherent skin-based marker errors due to marker misplacements (offset) and soft-tissue artifacts (STA). This paper aimed to assess the individual contribution of each of these errors to kinematic and kinetic shoulder outcomes computed using a shoulder rigid-body model. Baseline experimental data of three shoulder planar motions in a young healthy adult were collected. The baseline marker trajectories were then perturbed by simulating typically observed population-based offset and/or STA using a probabilistic Monte-Carlo approach. The perturbed trajectories were then used together with a shoulder rigid-body model to compute shoulder angles and moments and study their accuracy and variability against baseline. Each type of error was studied individually, as well as in combination. On average, shoulder kinematics varied by 3%, 6% and 7% due to offset, STA or combined errors, respectively. Shoulder kinetics varied by 11%, 27% and 28% due to offset, STA or combined errors, respectively. In conclusion, to reduce shoulder kinematic and kinetic errors, one should prioritise reducing STA as they have the largest error contribution compared to marker misplacements.

## 1. Introduction

Assessments of an individual’s shoulder kinematics and kinetics are critical to evaluate shoulder function [1], coordination [2], and to assess muscular forces [3]. As opposed to direct measurements, inverse kinematics and inverse dynamics [4] present advantages in computing accurate joint angles and moments as they utilise a global optimisation method [5]. However, the accuracy and reliability of the computed shoulder angles and moments entirely depend on the fidelity of the experimental motion capture data and on the underlying rigid-body upper-limb model.

Regarding the latter, personalisation of rigid-body models is commonly undertaken to match the studied individual’s anthropometric measurements and inertial properties [6,7,8]. This ensures that the inverse kinematic and inverse dynamic computations are based on a close representation of the person’s anatomy, and is essential to obtain physiologically consistent joint angles [9,10].

In order to experimentally record shoulder kinematics, skin-based marker systems are considered the gold-standard, thanks to their accessibility, non-invasive placement [11] and reliability [12]. However, the nature of skin-based markers presents two main issues, especially at the shoulder joint.

Firstly, palpating bony landmarks through soft tissues, such as skin, fat and muscles, can be challenging, particularly at the scapular and spinal levels. Even though shoulder landmarks have been well defined by the International Society of Biomechanics (ISB) [13], the reliability of their locations are dependent on the operator’s level of expertise [14]. Moreover, inter-operator variability from a few millimetres up to a centimetre has also been demonstrated [15]. Marker misplacement (offset) is undoubtedly responsible for errors in the computed kinematics and kinetics, but the extent to how much marker offset contributes to overall errors is still unknown.

Secondly, the most redundant issue with skin-based markers is their propensity to follow skin motion instead of the underlying bony landmark [16]. This issue is known as soft-tissue artifact (STA). STAs are common over the entire human body, although they are predominant at the spinal and scapular levels, which, in some cases, makes it impossible to accurately track a bony landmark. To limit this issue at the scapula, a scapula marker cluster is often used as described by Warner et al. [17,18]. This tool averages the STA to only one point of the scapula, i.e., the acromion, where soft tissues are the thinnest [19]. Moreover, studies have demonstrated the influence of STA on shoulder kinematic computations [20,21,22,23]. However, none of these studies has separately quantified the effects of STA from those of marker placement at the shoulder.

Myers et al. [9] used rigid-body modelling and probabilistic (i.e., Monte-Carlo) simulations to study the individual influence of offset and STA on lower-limb kinematics and kinetics during walking. They based their work on previously published directional measurements describing typical marker misplacements [24] and STA [25] observed in the lower-limbs when using marker-based systems. This modelling work is unique and essential to better understand how errors propagate through rigid-body computations. Ackland et al. [26] and Wu et al. [27] used similar methods to quantify the error propagation from kinematics, muscle moment arms and tendon properties to muscle and joint forces in the upper-limb using musculoskeletal modelling. However, no study has addressed the variability produced by experimental marker errors, i.e., marker misplacements and STAs, on shoulder kinematics and kinetics.

Studied individually, we hypothesise that STAs will have a greater impact on shoulder kinematics and kinetics than the effects of marker misplacements. Moreover, the greatest variability in shoulder kinematics and kinetics would occur when both error types are combined (Combined).

In order to verify these hypotheses, the present study will use a personalised upper-limb rigid-body model along with baseline experimental data of three shoulder planar motions, within which modelling and experimental biases will be carefully removed. Subsequently, a Monte-Carlo approach will be adopted simulating typically observed upper-limb marker offset and STA, taken from the literature. Each of these experimental marker errors will be studied individually, as well as in combination. This methodology aims to identify the individual effects, via the related accuracy and variability, produced by each type of experimental marker error (i.e., offset and STA) on shoulder kinematics and kinetics. Moreover, the accuracy and variability of the computed shoulder kinematics and kinetics will be studied in the case of combined simulated offset and STA to represent the effects of realistic experimental marker errors.

## 2. Methods

### 2.1. Participant

A healthy volunteer (female, age: 29 y.o., weight: 57.2 kg, height: 157.5 cm) with no history of shoulder pain or pathology was recruited for this study. Ethics approval was obtained from the QUT Human Research ethics committee, and the participant provided written informed consent before participation (QUT ethics approval number #2000000470).

The participant underwent an MRI scan of the upper arm and shoulder and participated in a motion capture recording of upper limb movements on separate days. Hemithorax and dominant upper extremity (including whole spine and sternum) were imaged using a 3 Tesla MR scanner (Ingenia, Koninklijke Philips N.V., Amsterdam, The Netherlands) at voxel sizes of 0.4 × 0.4 × 0.8 mm (using a T1 Dixon sequence). The sternum, clavicle, scapula and humerus in the images were then segmented using *Mimics 23.0* (Materialise, Leuven, Belgium) to create a 3D anatomical model.

### 2.2. Experimental Motion Trajectories

During the motion capture session, seven retro-reflective markers were attached to relevant anatomical landmarks of the thorax (C7, T8, IJ, PX), right shoulder (AC) and arm (EL, EM) according to the ISB standards [13] (cf. Figure 1). A marker cluster was placed on the flat bony part of the scapular spine, close to the acromion, and used to track scapula motions as well as average and limit skin motion artifacts [17,19]. Other scapular landmarks (AAc, TS, AI, PC) were then located in the scapular coordinate system using a registration wand [18]. Lastly, a marker cluster was attached to the participant’s dominant upper arm, and the participant was asked to perform shoulder circumduction for 30 s. This task allowed estimation of the location of the functional glenohumeral joint (GHJ) centre using the ScoRE method [28].

The participant was instructed to stand and perform a trial for three different shoulder planar motions, i.e., shoulder abduction/adduction (AA), flexion/extension (FE) and internal/external rotation (IER). AA and FE tasks were executed up to shoulder level (approximately 90° of elevation) to limit skin-motion artifacts at the scapula cluster [17]. IER was executed up to contact of the forearm with the thorax. Each task was performed with the arm along the thorax and following three consecutive phases, i.e., the initial movement, a holding phase and the opposite movement to come back to initial posture. A metronome directed each phase to last a period of 2 s. Marker trajectories were recorded at 200 Hz using a 12-camera motion capture system (Vicon Motion Systems, Oxford, UK) and filtered using a second-order, zero-lag, 4 Hz low-pass Butterworth filter [29].

### 2.3. Multi-Body Model

The model developed for the study was structured as a five segment, eight degrees-of-freedom (DOF) multi-body model of the upper limb. The sternoclavicular joint (SCJ) was modelled as a 2 DOF universal joint, and both the acromioclavicular joint (ACJ) and GHJ as 3 DOF spherical joints. The elbow and wrist joints were assumed to be fixed with no degree of freedom.

An anatomical model was designed from segmented MR images of the sternum, clavicle, scapula and humerus bones to create an image-based subject-specific model of the participant. Segmental parameters including centre of mass and moments of inertia were determined from the images [30]. The forearm and hand bones could not be retrieved from the MRI, thus generic shapes were used and linearly scaled to estimate the inertial properties of the forearm. In the following, the model is denoted as an MRI-based model (cf. Figure 1).

Twelve bony landmarks, described by the ISB [13] and belonging to the thorax, clavicle, scapula and humerus segments, were digitally selected on the virtual bony surfaces by an experienced operator using the open-source software *NMSbuilder* (cf. Figure 1) [30,31]. The model-based GHJ centre was determined as the centre of the best-fit sphere onto the humeral head [32]. Bony landmarks and joint coordinate systems were defined based on the ISB recommendations [13]. Based on Šenk et al., the Cardan sequence YXZ was used for each joint kinematic assessment to limit gimbal lock issues [13,33].

### 2.4. Baseline Trajectories

In order to individually study the influence of marker misplacements and STA, the following method was used to firstly remove any modelling bias out of the marker trajectories. From the experimental trajectories, the corresponding joint angles were computed using inverse kinematics, via *OpenSim 3.3* API in *MATLAB* and the MRI-based model. Then, via the Point Kinematics tool available in *OpenSim*, the exact trajectories of all the model’s surface landmarks were computed. Hence, these corrected trajectories do not contain any STA and perfectly represent the shoulder landmarks and their movements. From these baseline trajectories, errors can be artificially produced and their individual effects on shoulder kinematics and kinetics can be examined.

### 2.5. Marker Perturbation

As outlined by Myers et al. [9], the two different experimental perturbation types simulated were (1) realistic marker placement offset and (2) STA. Ultimately, a third perturbation type was simulated when both offset and STA, were combined, representing what realistically happens in a motion capture laboratory (see [9] for further explanations and illustrations).

#### 2.5.1. Marker Offset

The first simulated perturbation type was an offset in the marker placement with respect to its underlying bony landmark. Inter-operator palpation variability was taken from the literature [15]. The simulated marker offset, $r^{offset}$, was assumed constant throughout the motion in the local coordinate system of the segment and was randomly chosen from the normal distribution (ND): 0 ± SD. The standard deviation (SD) was dependent on the marker location and described in Table 1 [15]. The mean equal to 0 means that, on average, the marker would be correctly located at the accurate bony landmark position.

#### 2.5.2. STA

The STA simulation was undertaken by assuming that artifacts depended on the phase of the motion and the motion type. For five different motion phases, respectively, from 0 to 10, 10 to 33, 33 to 66, 66 to 90 and 90 to 100%, the STA of the thoracic, scapular and humeral markers, i.e., $r^{STA}$, were randomly selected from their different NDs, as reported in Table 2. Note that we selected five phases based on the available data from Konda et al. [34].

In detail, the AAc landmark was perturbed in each x-, y- and z-direction of the local scapular coordinate system, based on data from the literature [34] and as reported in Table 2. Unfortunately, no data were found regarding directional STA at the AAc landmark during an IER task. Therefore, similar values to those obtained during the AA task were assumed, based on van Andel et al. [35]. The other scapular landmarks such as TS, AI and PC followed the same errors as AAc to replicate the fact that these landmarks were registered using the scapular marker cluster placed on the acromion. Regarding the thoracic, clavicular and humeral landmarks, no directional information was found in the literature. Thus, STA for the 6 thoracic and humeral landmarks was assumed to follow the ND: 0 ± 4 mm, based on Konda et al. [34]. Continuity of every marker trajectory was assured between the five motion phases by locally weighted smoothing [36].

### 2.6. Monte-Carlo Approach

A Monte-Carlo approach was chosen to quantify the kinematic and kinetic variabilities due to both of the simulated experimental marker errors. This method allows statistical quantification of the plausible and realistic ranges of outcomes that would occur from a motion capture session, accounting for the known probabilistic distribution of each source of errors.

For every iteration of the Monte-Carlo simulations, each of the 12 baseline 3D trajectories, i.e., $X_{i,j}\left(t\right)$, $Y_{i,j}\left(t\right)$ and $Z_{i,j}\left(t\right)$, was randomly perturbed as described in Equations (1) to (3). i corresponds to marker 1 to 12, j corresponds to motion phase 1 to 5.
(1)$$X_{i,j}\left(t\right)=X_{i}\left(t\right)+r_{X,i,j}^{STA}+r_{X,i}^{offset}$$
(2)$$Y_{i,j}\left(t\right)=Y_{i}\left(t\right)+r_{Y,i,j}^{STA}+r_{Y,i}^{offset}$$
(3)$$Z_{i,j}\left(t\right)=Z_{i}\left(t\right)+r_{Z,i,j}^{STA}+r_{Z,i}^{offset}$$

The present study comprised three different Monte-Carlo simulations per motion type, each focusing on one of the three types of perturbations (i.e., offset, STA and Combined). When offset is simulated alone, all $r_{Y,i,j}^{STA}=0$. In contrast, if STA is simulated alone, all $r_{i}^{offset}=0$. Each simulation included 1000 iterations assuming convergence would be reached priorly.

### 2.7. Analysis

All perturbed trajectories were used, together with the MRI-based model, to perform inverse kinematics and inverse dynamics using *OpenSim 3.3* API in *MATLAB*. These resulted in GHJ angles and moments during the three different shoulder tasks. Only the main GHJ angle responsible for each task will be used and reported in the current study, i.e., X for AA, Z for FE and Y for IER.

The effects on kinematic and kinetic results due to marker misplacement and STA were analysed through data convergence, accuracy and variability by reporting the 50% median and the 5–95% bounds of the 1000 iterations per simulation, and compared to the baseline GHJ angles and moments.

First, convergence of each statistical simulation was checked and assumed satisfied if and when the convergence criterion ε(i) at the ith iteration, given by Equation (4), reached 5 × 10^−4^ or less. ε(i) represents the averaged changes in the last 10 iterations of the maximum difference, $\Delta_{5-95\%}^{max,\;i}$, between the 5 and 95% bounds over the duration of a trial. The number of iterations and the level of convergence were arbitrarily selected and were found to be adequate for this particular study.
(4)$$\varepsilon \left(i\right)=\frac{\sum_{j=i}^{i+9}\left|\frac{\Delta_{5-95\%}^{max,\;j+1}-\Delta_{5-95\%}^{max,\;j}}{\Delta_{5-95\%}^{max,\;j}}\right|}{10}$$

The accuracy of each perturbed simulation, to target the baseline, non-perturbed, kinematics and kinetics, was studied by calculating the Root Mean Square Error (RMSE) between the baseline angle/moment and the 50% median over time. 

Finally, variability obtained from each perturbed simulation was studied by calculating the difference between the 5 (Min) or 95% (Max) bound and the 50% median result, averaged over the duration of the normalised time.

## 3. Results

All kinematic simulations converged using between 120 (offset–AA task) and 323 (STA–FE task) iterations depending on the perturbation type and the motion task, with an average of 234 iterations. All kinetic results reached convergence between 109 (offset–AA task) and 397 (STA–FE task) iterations, with an average of 244 iterations.

Figure 2 presents the ranges of GHJ angles and moments computed out of the thousand iterations of each Monte-Carlo simulation (i.e., kinematics and kinetics of the three different shoulder tasks) with the medians, and the bounds containing 5–95% of all results over the normalised task time. On average, the kinematic and kinetic 5–95% bounds were larger during the hold phase for each task than while moving (cf. Figure 2). Moreover, one can generally observe that offset was responsible for less variability than STA alone. Combined offset and STA resulted in the largest variation.

The accuracy of the perturbed kinematics and kinetics is presented in Table 3 for the different tasks and perturbation types. The median of the perturbed kinematics was accurate on average with an RMSE of 0.79, 0.78 and 1.14° for the AA, FE and IER tasks, respectively. The median for the perturbed kinetic results was accurate on average with an RMSE of 0.09, 0.08 and 0.08 N·m for the AA, FE and IER tasks, respectively. 

Variabilities of the kinematic and kinetic results are presented in Table 4 for the different tasks and perturbation types. On average, kinematic variability (5–95% bounds) incrementally increased with each perturbation type, starting from offset, i.e., [−0.54, 1.14]°, STA, i.e., [−2.37, 1.91]°, to Combined, i.e., [−2.49, 2.13]°. Marker misplacements were responsible for 3% of the range of motion’s variability, STA for 6% and Combined for 7%. Similarly, kinetic variability incrementally increased with each perturbation type, starting from offset, i.e., [−0.25, 0.21] N·m, STA, i.e., [−0.45, 0.55] N·m, to Combined, i.e., [−0.52, 0.56] N·m, confirming what could be observed in Figure 2. Marker misplacements were responsible for 11% of the range of moment’s variability, STA for 27% and Combined for 28%. Note that, task-wise, kinematics varied by up to 4%, 12% and 4% of the whole range of motion for the AA, FE and IER tasks, respectively. Kinetics varied by up to 18%, 25% and 43% of the whole range of joint moments for the AA, FE and IER tasks, respectively.

## 4. Discussion

This study aimed to statistically quantify the individual and Combined effects of each experimental error due to skin-based markers, i.e., offset and STA, on shoulder kinematics and kinetics computed using a rigid-body model.

A few assumptions were considered. In the current study, directional uncertainties for the humeral and thoracic landmarks had to be assumed. To the best of our knowledge, these values are currently lacking in the literature. Although skin marker errors have been extensively studied [37], they are usually investigated from the point of view of their effects (e.g., kinematic errors produced from STA), but not their natures. Very rarely are mentioned values about directional biases (i.e., values in the spatial directions), which cause the kinematic and kinetic variabilities. We believe that the present study for the first time linked the experimental marker error’s natures and effects. 

Another limitation arose from the fact that STA are not only dependent on the body location, the motion task and the phase of the motion, but they are also subject-specific [20] and are closely linked to BMI [38,39].

Moreover, in the present study, the GH joint virtual marker was not perturbed, as it is not a skin marker per se. Experimentally locating the GH joint is usually undertaken by estimation techniques, which can either be geometric, regressive [32] or functional [28,40] and therefore can present high uncertainty as well. If properly accounted for, we could foresee that uncertainty on the GH joint location would increase the kinematic and kinetic variabilities to be expected during a standard marker-based motion capture session.

The study of a single subject, whose image-based model and experimental marker trajectories were used, was believed to be sufficient and to provide results that can be generalised, as the perturbed errors were selected from population-based data, i.e., *N* = 5 for marker misplacements [15] and *N* = 19 for STA [34]. 

Nonetheless, the same protocol cited herein was also applied to a taller, bigger, younger male subject (23 y.o., 178 cm, 77.5 kg, provided written consent and following the previously cited ethics approval number) in order to verify this assumption and have anthropometric variation. The acquired results showed similar trends, convergence, accuracies and variabilities of the shoulder kinematics and kinetics with respect to perturbed marker trajectories. Thus, we are confident that the reported variabilities were exclusively produced by the simulated marker errors and that modelling bias and anatomical variation only played a minor, negligible role in the present study.

Furthermore, studies that report shoulder kinematic errors caused by skin markers commonly studied the combination of errors, i.e., marker misplacement and STA as one parameter. Hamming et al. [39] and Lempereur et al. [41] both reported a RMSE between 2 and 6° for a shoulder elevation task. Van Andel et al. [35] reported a RMSE between 1 and 9° degrees. Our results from the combined marker errors are in agreement with these previous data (see Table 4). 

The reported kinematic and kinetic variabilities should be considered as confidence intervals for future IK and ID outcomes using a shoulder model. Moreover, the variabilities due to STA were on average two times larger than the one due to misplacements. Thus, to limit kinematic or kinetic errors, one should prioritise reducing STA before marker misplacements.

Wu et al. [27] undertook a Monte-Carlo approach applied to musculoskeletal modelling to study how computed muscle forces and joint forces varied with respect to perturbed kinematics. Our study is complementary to the study by Wu et al. as it addresses a step prior to the kinematic analysis, i.e., how errors in the experimental marker trajectories perturb the kinematics and kinetics.

Finally, the accuracy shown in Table 3 suggested that, theoretically, if enough experimental protocols were repeated, the median of all values will be representative of the real kinematics and kinetics of the shoulder joint despite marker misplacement and STA. Note also that variabilities reported in Table 4 for the simulated offset and STA didn’t add up to equal these of the Combined case, suggesting that marker offset distributions might have overlapped the distribution of STA, or vice versa.

## 5. Conclusions

This study used a probabilistic approach to investigate the individual effects of skin marker misplacements and STA on shoulder kinematics and kinetics and inform on the sensitivity of inverse kinematic and kinetic computations using shoulder rigid-body models. It was demonstrated that, during shoulder motion, STA contributed to greater kinematic and kinetic errors than marker misplacements, on average by two-fold. It was proposed that, to limit kinematic or kinetic errors, one should prioritise reducing STA before marker misplacements.

## Figures and Tables

**Figure 1 life-12-00819-f001:**
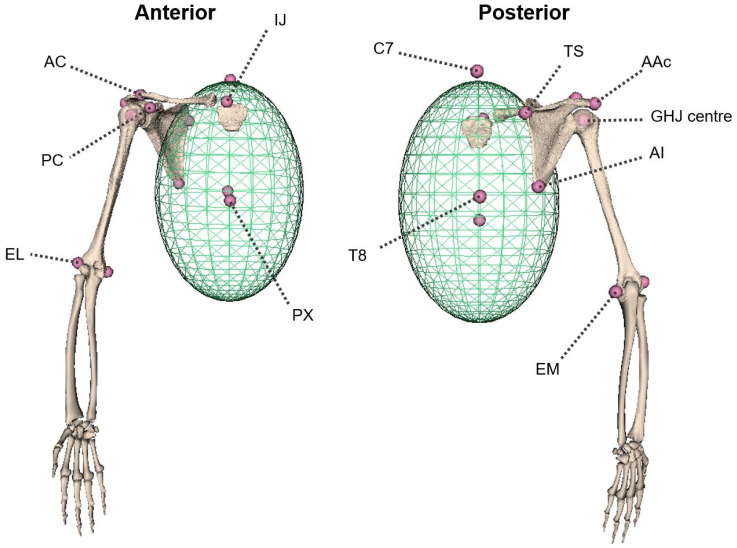
MRI-based model of the right shoulder and its bony landmarks. IJ: Incisura Jugularis, PX: Processus Xiphoideus, C7: 7th cervical vertebrae, T8: 8th thoracic vertebrae, AC: Acromioclavicular joint, AAc: Angulus Acromialis, TS: Trigonum Spinae Scapulae, AI: Angulus Inferior, PC: Processus Coracoideus, EM: Medial Epicondyle, EL: Lateral Epicondyle.

**Figure 2 life-12-00819-f002:**
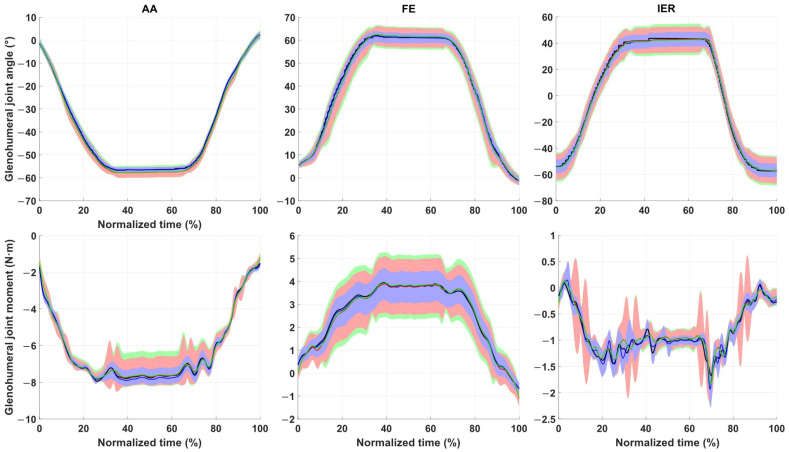
Kinematic (**top** row) and kinetic (**bottom** row) variabilities due to marker offset, Soft-Tissue Artifacts and Combined, respectively, in blue, red and green, during Abduction/Adduction (AA) (**left**), Flexion/Extension (FE) (**middle**) and Internal/External Rotation (IER) (**right**) tasks. Baseline kinematics and kinetics are represented using a black line. Coloured lines represent the median (i.e., 50% bound), shaded areas represent the 5–95% bounds.

**Table 1 life-12-00819-t001:** Standard deviation describing the inter-operator variability in palpating the accurate bony landmarks of the upper limb and thorax. Values described in their segment local coordinate system. Taken from de Groot et al. [15].

Skeletal Landmarks	Abbreviations	SD–X (mm)	SD–Y (mm)	SD–Z (mm)
Incisura Jugularis	IJ	1.4	1.6	1.9
Processus Xiphoideus	PX	2.1	1.6	2.2
7th cervical vertebrae	C7	2.3	3.3	1.9
8th thoracic vertebrae	T8	1.3	1.2	3.1
Acromioclavicular joint	AC	2.3	1.0	2.7
Angulus Acromialis	AAc	2.9	1.6	3.2
Trigonum Spinae Scapulae	TS	3.8	2.0	2.5
Angulus Inferior	AI	3.8	1.8	3.0
Processus Coracoideus	PC	2.3	1.0	2.7
Medial epicondyle	EM	1.8	2.3	1.8
Lateral epicondyle	EL	1.8	2.3	1.8

**Table 2 life-12-00819-t002:** Normal distributions (ND) used to perturb each bony landmark during each phase and function of the motion task. Values are described in their segment local coordinate system. Scapular values are taken from Konda et al. [34]. Note that soft-tissue artifacts for the six thoracic and humeral landmarks were arbitrarily assumed to follow the ND: 0 ± 4 mm.

Skeletal Landmarks	Motion	Simulated Phases	ND in Local X (mm)	ND in Local Y (mm)	ND in Local Z (mm)
AAc (TS, AI, PC followed the same NDs)	AA, IER	Phases 1 & 5	0 ± 0	0 ± 0	0 ± 0
		Phases 2 & 4	6.4 ± 4.1	3.6 ± 2.9	0.7 ± 7.5
		Phase 3	8.4 ± 4.5	6.0 ± 2.9	−0.8 ± 6.3
AAc (TS, AI, PC followed the same NDs)	FE	Phases 1 & 5	0 ± 0	0 ± 0	0 ± 0
		Phases 2 & 4	10.8 ± 4.8	3.7 ± 1.7	13.9 ± 7.5
		Phase 3	16.0 ± 5.0	7.9 ± 3.6	11.9 ± 7.8
IJ, PX, C7, T8, AC, EL, EM	AA, FE, IER	Phases 1, 2, 3, 4, 5	0 ± 4	0 ± 4	0 ± 4

**Table 3 life-12-00819-t003:** Accuracy of the median to estimate baseline kinematics and kinetics—RMSE calculated between the baseline angle/moment and the median result, over time, and for each task type and each perturbation type.

	RMSE Between Baseline and Median	Offset	STA	Combined
Abduction/Adduction	Angles (°)	0.18	1.17	1.03
	Moments (N·m)	0.08	0.10	0.11
Flexion/Extension	Angles (°)	0.40	0.98	0.92
	Moments (N·m)	0.06	0.10	0.09
Internal/External Rotation	Angles (°)	0.93	1.24	1.26
	Moments (N·m)	0.07	0.09	0.09

**Table 4 life-12-00819-t004:** Variabilities of the perturbed kinematics/kinetics compared to median—Difference between the 5 (Min) or 95 (Max) bound and the median, averaged over time, for each task type and each perturbation type.

	Averaged Difference Between Min/Max Bound and Median		Offset	STA	Combined
Abduction/Adduction	Angles (°)	Min	−0.23	−1.06	−0.99
		Max	+0.60	+1.12	+1.40
	Moments (N·m)	Min	−0.16	−0.41	−0.47
		Max	+0.21	+0.72	+0.74
Flexion/Extension	Angles (°)	Min	−1.13	−3.27	−3.92
		Max	+1.98	+3.12	+3.77
	Moments (N·m)	Min	−0.34	−0.77	−0.92
		Max	+0.22	+0.26	+0.31
Internal/External Rotation	Angles (°)	Min	−0.26	−2.78	−2.55
		Max	+0.83	+1.48	+1.23
	Moments (N·m)	Min	−0.07	−0.18	−0.17
		Max	+0.21	+0.67	0.63

## Data Availability

The data presented in this study are available on request from the corresponding author.

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
