# Peer review of "Statistical Quantification of the Effects of Marker Misplacement and Soft-Tissue Artifact on Shoulder Kinematics and Kinetics"

_life, 2022, doi:10.3390/life12060819_

Round 1

Reviewer 1 Report

Overall this research is very interesting and was conducted in a well-organized manner. The paper is well-written; a good read. However, there are some weak spots that either need a more elaborate clarification or a better justification of choices.

This study investigates the individual and combined effects of marker misplacements and soft-tissue artifacts in order to reduce shoulder kinematic and kinetic errors using a shoulder rigid-body model. Authors collected the data of three shoulder motions in one young participant. 

Line 148: under which circumstances the GHJ center was determined?
Line 150: The authors should elaborate more about The Cardan Sequence YXZ.

Line 183: how did authors come to the five phases of motions alongside with motion types?

Line 213: The 1000 iterations were assumed for convergence until reached priorly. How did authors decided on this number? 

Line 269: What does ROM stand for in estimating variability?

I have one final concern with the discussion section; authors should highlight the discussion section and focus on the implications of their research rather than focusing on the limitations of their user study. Because the current section lacks vision with the inability to summarize the authors' point of view.

I suggest authors to have conclusion section separately from the discussion. Also I suggest rewriting the conclusion it seems to arbitrary.

Author Response

We would like to thank the reviewer for the provided comments. We have addressed all questions and suggestions (see below) and believe that our updated work has been significantly improved upon the previous version thanks to the reviewer.

  • "Line 148: under which circumstances the GHJ center was determined?"

In line 148, the model-based GHJ centre refers to an anatomical feature in the MRI-based model (cf. Wu et al., 2005), and was determined as the centre of the spherical portion of the humeral head (cf. Meskers et al., 1998).

Note that it is different from the functional GHJ centre described between lines 114 and 116, and determined using a functional estimation method.

In the manuscript, both types of GHJ centres are well-described and named accordingly, i.e., model-based vs functional GHJ centres.

  • "Line 150: The authors should elaborate more about The Cardan Sequence YXZ."

The Cardan sequence was selected for kinematic computation to limit gimbal lock issues as suggested by Šenk et al., 2006. This information has been added to the manuscript as stated below (or lines 150-151 in the manuscript)

“Based on Šenk et al., the Cardan sequence YXZ was used for each joint kinematic assessment to limit gimbal lock issues”

  • "Line 183: how did authors come to the five phases of motions alongside with motion types?"

The three motion types, i.e., Abduction-Adduction, Flexion-Extension and Internal-External rotation, were selected as they are simple, planar tasks, for which we could likely find data about their soft-tissue artifacts in the literature.

Similarly, the five phases of motion came from Konda et al, 2018 who reported soft-tissue artifacts at 0, 30 and 60degrees of shoulder elevations. During move-hold-move motions like the ones we studied, it naturally corresponded to the 5 phases of motion, i.e., 0 to 10%, 10 to 33%, 33 to 66%, 66 to 90% and 90 to 100%.

We added the following statement to lines 185-186:

“Note that we selected five phases based on the available data from Konda et al. [34].”

  • "Line 213: The 1000 iterations were assumed for convergence until reached priorly. How did authors decided on this number? "

This value was determined arbitrarily. We then checked using the convergence criteria that our simulations converged. It appeared that they converged before the 1000 iterations, on average around 234 iterations, as mentioned in line 241.

We added this statement to the manuscript, lines 229-230:

“The number of iterations and the level of convergence were arbitrarily selected”

  • "Line 269: What does ROM stand for in estimating variability?"

ROM stood for “range of motion”. Although, we decided to remove the acronym and keep the full term throughout the document.

  • “I have one final concern with the discussion section; authors should highlight the discussion section and focus on the implications of their research rather than focusing on the limitations of their user study. Because the current section lacks vision with the inability to summarize the authors' point of view.”

We added a paragraph in the discussion emphasizing the principal take-home messages of the paper. See below (or between lines 319 and 323 in the manuscript):

“The reported kinematic and kinetic variabilities should be considered as confidence intervals for future IK and ID outcomes using a shoulder model. Moreover, the variabilities due to STA were on average two times larger than the one due to misplacements. Thus, to limit kinematic or kinetic errors, one should prioritise reducing STA than reducing marker misplacements.”

  • “I suggest authors to have conclusion section separately from the discussion. Also I suggest rewriting the conclusion it seems to arbitrary.”

We added a conclusion section and expanded it with the main findings of the manuscript and the principal discussion point. See below (or between lines 337 and 343 in the manuscript).

“This study used a probabilistic approach to investigate the individual effects of skin marker misplacements and STA on shoulder kinematics and kinetics and inform on the sensitivity of inverse kinematic and kinetic computations using shoulder rigid-body models. It was demonstrated that, during shoulder motion, STA contributed to greater kinematic and kinetic errors than marker misplacements, on average by two-fold. It was proposed that to limit kinematic or kinetic errors, one should prioritise reducing STA than reducing marker misplacements.”

References:

Wu, G.; Van Der Helm, F. C. T.; Veeger, H. E. J.; Makhsous, M.; Van Roy, P.; Anglin, C.; Nagels, J.; Karduna, A. R.; McQuade, K.; Wang, X.; Werner, F. W.; Buchholz, B. ISB Recommendation on Definitions of Joint Coordinate Systems of Various Joints for the Reporting of Human Joint Motion - Part II: Shoulder, Elbow, Wrist and Hand. Journal of Biomechanics 2005, 38 (5), 981–992.

Meskers, C. G. M.; Helm, F. C. T. Van Der; Rozendaal, L. A.; Rozing, P. M. In Vivo Estimation of the Glenohumeral Joint Rotation Center from Scapular Bony Landmarks by Linear Regression. Journal of Biomechanics 1998, 31 (1), 93–96.

Šenk, M.; Chèze, L. Rotation Sequence as an Important Factor in Shoulder Kinematics. Clinical Biomechanics 2006, 21 (SUPPL. 1), 3–8.

Reviewer 2 Report

This manuscript described the effects of marker misplacements (Offset) and soft-tissue artifacts (STA) on shoulder kinematics and kinetics using Monte-Carlo approach.

The methods and results are clearly presented.

The only main concern I have is the sample size used in this study. Although authors elaborate single subject study in Line 300-303. Considering References 27 and 34, I think this manuscript may be more suitable as Case Reports by Life.

Author Response

We would like to thank the reviewer for the provided comments.

We understand why the reviewer is concerned about the sample size of the study. However, this justifies why we have spent a large section of the discussion (between lines 301 and 312) explaining why the results can be generalisable.

Between lines 305 and 312, we explained that we ran the same simulation using another participant, with different demographic and anthropometric parameters than the participant of the study. We got similar results.

Moreover, this generality likely comes from the fact that the normal distributions used to simulate marker errors were determined from a whole population, i.e., representative of a whole cohort and not specific to a single participant. This was explained between lines 301 and 304.

Reviewer 3 Report

Thank you for inviting me to review Statistical Quantification of the effects of marker 2 malpositioning and soft tissue Shoulder artifacts 3 Kinematics and kinetics 4. By this means I send comments on the article. The work in general is interesting, because it analyzes the errors of two techniques, observations: 1.- Show in the conclusion if the hypotheses raised were met with line 82 of
the manuscript. 2.-It does not mention the simulation times according to line 238,
nor the total of this time each interaction, is this time important??? 3.- Lia 267 which is the ROM, it does not define it, it is not important?? 4.-Expand conclusion compared to what was explained in the discussion? 6.- Is it possible to activate bibliographical references referring to the topic????

Author Response

We would like to thank the reviewer for the provided comments. We have addressed all questions and suggestions (see below) and believe that our updated work has been significantly improved upon the previous version thanks to the reviewer.

  • “Show in the conclusion if the hypotheses raised were met with line 82 of
    the manuscript.”

We added a conclusion section and expanded it with the main findings of the manuscript. We believe it now confirms the hypothesis stated line 82. See below (or between lines 337 and 343 in the manuscript).

This study used a probabilistic approach to investigate the individual effects of skin marker misplacements and STA on shoulder kinematics and kinetics and inform on the sensitivity of inverse kinematic and kinetic computations using shoulder rigid-body models. It was demonstrated that, during shoulder motion, STA contributed to greater kinematic and kinetic errors than marker misplacements, on average by two-fold. It was proposed that to limit kinematic or kinetic errors, one should prioritise reducing STA before marker misplacements.”

  • “It does not mention the simulation times according to line 238,
    nor the total of this time each interaction, is this time important???”

We believe that the computation time is not an important feature of the study. It does not bring anything new to the discussion or conclusion. For your reference, the simulation of a 1000 iterations took approximately five hours using Matlab and a computer with Intel(R) Core(TM) i7-8700 CPU @ 3.20GHz and 16GB RAM.

  • “Lia 267 which is the ROM, it does not define it, it is not important??”

ROM stood for “range of motion”. Although, we decided to remove the acronym and keep the full term throughout the manuscript.

  • “Expand conclusion compared to what was explained in the discussion?”

The main discussion point, stated below, was added to the conclusion. (lines 341 - 343 in the manuscript)

It was proposed that, to limit kinematic or kinetic errors, one should prioritise reducing STA before marker misplacements.”

  • “Is it possible to activate bibliographical references referring to the topic????”

Regarding bibliographical references, we followed the Life/MMDP guidelines.